# Structural Characteristics of the Main Resinous Stands from Southern Carpathians, Romania

Gabriel Murariu [1], Lucian Dinca [2], Nicu Tudose [2], Vlad Crisan [2,*], Lucian Georgescu [1], Dan Munteanu [3,*], Mihai Daniel Dragu [1], Bogdan Rosu [3] and George Dănuţ Mocanu [4]

[1] Chemistry, Physics, and Environment Department, Faculty of Sciences and Environment, "Dunărea de Jos" University of Galaţi, No. 111 Street Domnească, 800201 Galaţi, Romania; gabriel.murariu@ugal.ro (G.M.); lucian.georgescu@ugal.ro (L.G.); dragumihaidaniel@gmail.com (M.D.D.)

[2] "Marin Drăcea" National Institute for Research and Development in Forestry, 13 Closca Street, 500040 Brasov, Romania; dinka.lucian@gmail.com (L.D.); cntudose@yahoo.com (N.T.)

[3] Faculty of Automation, Computer Sciences, Electronics and Electrical Engineering, "Dunărea de Jos" University of Galaţi, No. 111 Street Domnească, 800201 Galaţi, Romania; Bogdan.rosu@ugal.ro

[4] Faculty of Physical Education and Sport, "Dunărea de Jos" University of Galaţi, No. 111 Street Domnească, 800201 Galaţi, Romania; george.mocanu@ugal.ro

\* Correspondence: vlad_crsn@yahoo.com (V.C.); dan.munteanu@ugal.ro (D.M.); Tel.: +40-744-688-968 (V.C.)

**Abstract:** The purpose of this study, which contains historical data recorded over a period of 40 years, was to identify the main factors that influence and control the level of wood mass production. The main reason was to optimize the management of forest areas and was driven by the necessity to identify factors that can influence most of the volume produced by coniferous forests located in southeast Europe. The data was collected between 1980 and 2005 at the National Institute for Research and Development in Forestry, for forests located in the Southern Carpathians, Romania. The studied data refer to the parameters that model forest structure for spruce, fir, pine, and larch. These are the main resinous species found in the Southern Carpathians. The total area covered by these forests is 143,431 ha. At the forest species level, the analysis consists of 16,162 records (corresponding to the elements of the trees), covering an area of 45,008 ha for fir, 4711 ha for larch, 81,995 ha for spruce, and 11,717 ha for pine. The aim of this research has been to investigate and to assess the impact and magnitude of abiotic factors such as altitude and field aspect on forest structures from the main resinous stands located in the Southern Carpathians. Taking into account the size of the database as well as the duration for collecting data, a complete statistical and systematic approach was considered optimum. This resulted from our wish to emphasize and evaluate the influence of each analysed factor on the wood mass production level. The relationship between abiotic factors and forest structure has been analysed by using a systematic statistical approach in order to provide a useful theoretical reference for the improvement of forest management practices in the context of multiple climatic, environmental, and socio-economic challenges. These common characteristics have been found by applying ANOVA and multivariate statistical methods such as PCA and FA methods. A series of parameters were considered in this investigation, namely altitude (ALT), forest site type (TS), forest type (TP), consistency (CONS) etc. In order to obtain a complete image, we have also applied multivariate analysis methods that emphasize the effect size for each database parameter. At such a level of recorded data, the statistical approach ensures a factor level of $p < 0.001$ while the accuracy in evaluating effect size is increased. As such, they influence the spreading and structure of the studied resinous stands to a higher degree, regardless of species.

**Keywords:** structure; spruce; fir; pine; larch; statistical approach

## 1. Introduction

Modelling forest structure has multiple purposes: upgrading inventory, assessing alternative silvicultural practices, and designing future management plans [1], as well as

assessing the relationship between forest structure and productivity [2] in order to guide and assist decision-makers in achieving sustainable management in accordance with their goals [3]. This activity requires empirical information related both to thinning and natural stand growth [4] and relies on an accurate description of forest dynamics [5].

Nevertheless, forest structure is greatly influenced by abiotic factors whose influence can be seen in forest composition and density being an integrated part of the forest ecosystem. Abiotic factors can have both positive and negative effects. The negative (disturbance) effect arises when one of the abiotic factors goes beyond the normal limit of forest habitat suitability [6–9].

In Romania, forests cover approximately 28% of the land, with almost 60% of these forests being located in the mountain region. The Carpathian Mountains represent a complex mountain system running from the central part of Europe, corresponding to Czech Republic territory, to the southeast, located in Romanian territory. The Carpathians are distributed from the Czech Republic, Slovakia, Poland, and Hungary in the west to Ukraine, Romania, and Serbia in the southeast. In Romania, the Carpathians are split into three groups: Eastern Carpathians, Western Carpathians, and Southern Carpathians.

Norway spruce (*Picea abies*) is the dominant native forest species in the Romanian Carpathians at elevations between 1200 and 1800 m, and is also one of the most important commercial tree species. Together with silver fir (*Abies alba*), pine (*Pinus*), and larch (*Larix deciduas*), this species forms 30% of the total Romanian forest cover [10], forming old-growth stands [11], and even 'smart forests' under some conditions [12]. Spruce has a protective role in areas with high slopesprone to erosion from the Southern Carpathians [13] or in areas with natural lake slopes [14]. Norway spruce (*Picea abies* (L.) H. Karst.) grows in the Dinaric Alps, the Alps, the Sudetes, and the Carpathians and throughout much of Scandinavia as well as in the lowlands of north-eastern Europe [15]. Norway spruce (*P. abies*) is a boreal-mountain species, characterised by a wide and fragmented distribution range. Silver fir (*A. alba* Mill.) is one of the most valuable resinous trees in Europe for both historical and economic reasons. Its relevance for Central European forestry is far greater than its proportion in forest stands, but silver fir is also very sensitive to silvicultural interventions [16]. Silver fir grows mainly in the mountainous regions of Europe, from the Pyrenees in the west to Central Poland and Germany in the north, the Alps and the Carpathians in the east, and Italy and Greece in the south and it often occurs at relatively high altitudes (500–2000 m a.s.l.). However, it is also distributed in lowlands, for instance in France, Poland and Ukraine [17].

In contrast to spruce and silver fir, pine species are much less frequent in Romania. The most common pine species are: *P. sylvestris* in low lands and knolls (mostly in plantations) and in the upper mountain region; *P. cembra*, restricted to small populations in a few mountain massifs, particularly in the north-eastern (Calimani Mountains) and southern Carpathians (Retezat Mountains); and *P. nigra*, which grows in the Banat Mountains of south-western Romania at elevations between 500 and 1200 m [18]. *P. mugo* is a shrub that is widespread in the sub-alpine belt (1600–2250 m) of the Carpathians [18]. Pine has also been used for degraded land afforestation [19,20].

In Romania, larch (*Larix decidua* Mill.) has a naturally discontinuous range, concentrated into five genetic centres that are named after their respective mountain massifs: Ceahlãu, Ciucas, Bucegi, Lotru and Apuseni [21]. In the Romanian Carpathians, as well as in the Alps, larch is located at the upper limit of vegetation, where it forms pure stands or mixed stands with spruce, Swiss stone pine, or even with fir, beech, ash, and birch. The total area of natural larch stands is around 4500 ha [22,23], representing 0.3% of the national forest area, but the species has expanded far beyond its natural area of vegetation. In 2007, the area of artificial larch stands amounted to 12,500 ha, according to the Romanian National Institute of Statistics. Ecologically, Romanian larch is a high-altitude species with high light requirements, and it grows on skeletal soils, limestone, and conglomerates [23].

The aim of this research has been to assess the impact of abiotic factors on forest structures from the main resinous stands located in the Southern Carpathians. The rela-

tionship between abiotic factors and forest structure has been analysed in order to provide a useful theoretical reference for the transformation of forest management practices in the context of multiple climatic, environmental, and socio-economic challenges. A systematic statistical approach was developed in order to reach the influence magnitude of these abiotic and forest structures on their volume considering the consistency of the records' database. A coherent statistical analysis can only be performed in the case of consistent databases with a large number of records. In this sense, the purpose was to identify, investigate, and prioritize the magnitude of the influence of external abiotic factors, in order to better manage the forest fund. The identification and ranking of the most important abiotic factors using a database covering a period of over 25 years is an original aspect, less presented in the literature. Another objective that was pursued was to demonstrate the possibility of analysis using multivariate statistical methods. For this purpose, the PCA (main component analysis) method and the FA (factor analysis) method were used complementarily. Therefore, on the one hand, the PCA method is based on the use of correlation factors to determine the relative angles between the representative vectors in order to construct a complete system of reference axes that minimize the dataset variant. On the other hand, the FA method is based on identifying system rotation transformations of natural axes so as to minimize distances to the system axes. Complementarily used methods are common in the literature. For the sake of a complete analysis, multivariate investigation methods were considered in order to obtain the size effect order. Last but not least, methods of topological statistical analysis were applied, such as cluster analysis.

## 2. Materials and Methods

The material for this article has been collected from 39 forest management plans [24] realised during 1980–2005 for forests located in the Southern Carpathians, Romania. The total area covered by the analysed forest management plans is 143,431 ha. At the forest species level, the analysis consists of 16,162 records (corresponding to the trees), covering an area of 45,008 ha for fir, 4711 ha for larch, 81,995 ha for spruce, and 11,717 ha for pine.

The Romanian Southern Carpathians area was chosen for this study. The Southern Carpathians are composed of a number of well-differentiated mountain massifs: Tarcu Mountains, Godeanu Mountains, Retezat Mountains, Latoriţa Mountains, Sureanu Mountains, Cindrel Mountains, Parang Mountains, Capatanii Mountains, Lotru Mountains, Fagaras Mountains, Bucegi Mountains, Iezer–Păpuşa Mountains, Leaota Mountains, Piatra Craiului Mountains, and some smaller ranges (Figure 1, marked with yellow).

We took into consideration only stands with 100% Norway spruce composition. For silver fir, larch, and pine, we used all the stands in our analysis, even the ones in which these species were poorly represented (10%). The reason for this choice is the very small distribution of pure larch and silver fir stands. In general, these species are typically spread out and contribute tothe composition of other tree stands.

The average diameter at breast height was measured for each stand by using a tree calliper, as follows: 2 to 4 trees for every two to three sampling plots in stands with ages of up to 60 years old; 3 to 5 trees in stands over 60 years, and 5 to 8 trees in uneven-aged stands for every three to five sampling plots.

The average height, with a tolerance of $+/- 10\%$, was measured for the representative trees from each age category. The age was calculated for each stand, with a tolerance of $+/- 10\%$, based on the forest management plan. In this sense, the calculation was based on the year when the regeneration occurred, the number of annual rings from the fresh tree stumps, and the number of years required to achieve the height of the stem at which the determination was made. Moreover, the age of each stand has been rounded up every 5 years for high-forest-regenerated stands and every 2 to 3 years for coppice-regenerated stands. The ages of the stands' trees species, which are part of management plans, are noted in this paper under the acronym 'VRT' (years).

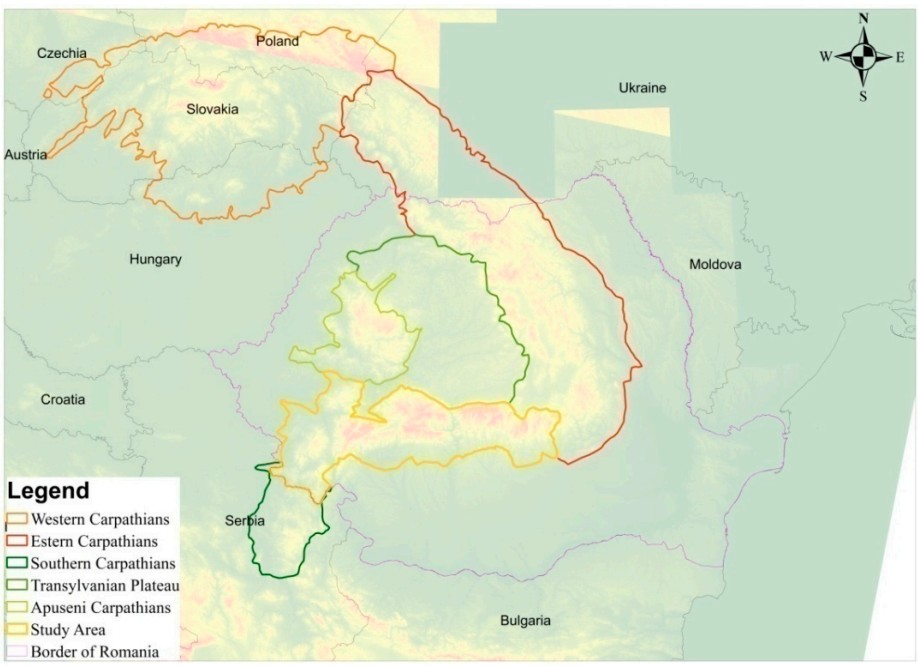

**Figure 1.** Study area.

Regarding wood volume for each stand, we have applied the following principles: the volume was calculated by statistical or integral inventories for stands that will be harvested in the first 10 years, with a tolerance of $+/-10\%$ at a confidence level of 90%; for the rest of the stands, the volume was calculated by simplified procedures (based on simplified survey-based production tables), with a tolerance of $+/-13\%$ at a confidence level of 90%. Hereafter, volume is denoted by 'VOL' and represents tree volume (cubic meters) on the surface unit (hectares).

The descriptions of all forest site parameters have been taken from forest management plans. Site descriptions have been based on soil studies, physical-geographic data, rock type, flora characteristics, and forest type. Soil types were denoted by 'SOL', field aspect by 'EXP', relief unit (field, mountain etc.) by 'RLF', field slope or inclination by 'INC', field configuration by 'CNF', canopy cover by 'CNS' (the proportion of a horizontal projection occupied by canopy, a number between 0.1 and 1), altitude by 'ALT', stand structure (which can be even-aged, relatively even-aged, uneven-aged, or relatively uneven-aged) by 'STR', forest type description (based on composition, origin, productivity etc.) by 'TP', station type by 'TS', and flora characteristics (species) present in the stands by 'FLR'.

In this paper, the data series set contained 12,376 records lines. The database proved to be consistent and lacked contradictory recordings as all measurements were validated before they were introduced in the dataset.

*Work Hypothesis*

**Hypothesis 1 (H1).** *The main factors that influence the development and value of wood production can be identified for different resinous species present in southeast Europe, namely in the Southern Carpathians.*

**Hypothesis 2 (H2).** *Significant differences exist between the factors that influence the development of different resinous species present in southeast Europe, namely in the Carpathians.*

In this way, a number of analysis methods were considered [25,26]. We followed a statistical approach similar to that of Popa et al., (2018) [27] by using the set of statistical analysis stages exposed in the indicated work. As such, we were able to follow and emphasize the way in which databases were analysed in stages.

In our approach, we have relied less on classical analysis (regressions) and more on the use of multivariate analysis methods of complementary results, such as factor analysis (FA) and principal component analysis (PCA). Because our datasets proved to be extremely consistent (over 12,000 records), we were able to investigate and evaluate the magnitude of the similarity between the analysed parameters by using cluster analysis methods. The data analyses were performed with STATISTICA 13.5.0.17 software and SPSS tools.

The Kolmogorov–Smirnov test was used for testing the normal distribution of the experimental dataset. Here again, we followed a statistical approach similar to that of Popa et al., (2018) [27].

The evaluation of the Pearson correlation coefficients was included in a preliminary stage in the data analysis program. Moreover, in order to make a complete and unitary evaluation of the dataset, we included alternative procedures in the group of study methods. Thus, for complementarity, the correlation coefficients were also analysed by comparison with results obtained through the stepwise regression method. The Lilliefors test is a normality test based on the Kolmogorov–Smirnov test. It is used to test the null hypothesis and, at the same time, to show that data comes from a normally distributed population. This test consists of adjusting the critical values of the Kolmogorov–Smirnov test [28].

The data analysis was first descriptive and in the first phase consisted of box plots or histograms containing the results of the Kolmogorov–Smirnov analysis. Moreover, in order to highlight the intensity of the correlations, we used a general linear/nonlinear model.

This Wald-type test evaluates the constraints on statistical parameters based on the weighted distance between the unrestricted estimate and the value of its hypothesis in the null hypothesis, where the weight is the accuracy of the estimate. Intuitively, the greater this weighted distance is, the less the constraint is true. Dataset variance and confirmation of strong links between measured and recorded parameters were analysed by using classical methods from the literature such as PCA method, FA, and similarity analysis that uses the cluster analysis method. Conclusions based on these analyses are similar and provide a complete picture of the used records set.

The FA method is often used in order to identify the main factor groups that affect the total dataset variance [29]. It is preferable to use the correlation matrix and not the covariance matrix because standardisation may cause distortions in the results as it changes deviations [27].

The PCA method was used to verify and to identify the factors that affect database variability [30,31]. The obtained results can sometimes differ between the PCA method and the FA method [27].

The cluster method is a method of analysing the degree of similarity, considering the datasets as representative points in multi-dimensional spaces. As such, similarity is evaluated according to the distance in chosen metrics. In this procedure, we chose the Euclidean metric of square type with complete connection, using weighted pair group averages [32].

In order to highlight and represent possible differences between species, we considered useful the graphical representation of 3D shape plots. Thus, we were able to highlight the dependencies of recorded volume values according to consistency, age, and altitude, considering the circular permutations of two of those parameters that proved to be sensitive for volume values. In this sense, we performed a representation of the volume dependence of consistency and elevation, consistency and age, etc. for each species, resulting in interesting details.

### 3. Results

*3.1. The Variation of Some Structure Parameters for the Main Resinous Species Located in Southern Carpathians, Romania*

Amongst the four resinous species from the Southern Carpathians, pine is situated at the lowest altitudes (with an average of 400–800 m), followed by larch (600–1400 m), which has the largest distribution range, fir (900–1300 m), and spruce (1200–1800 m) (Figure 2).

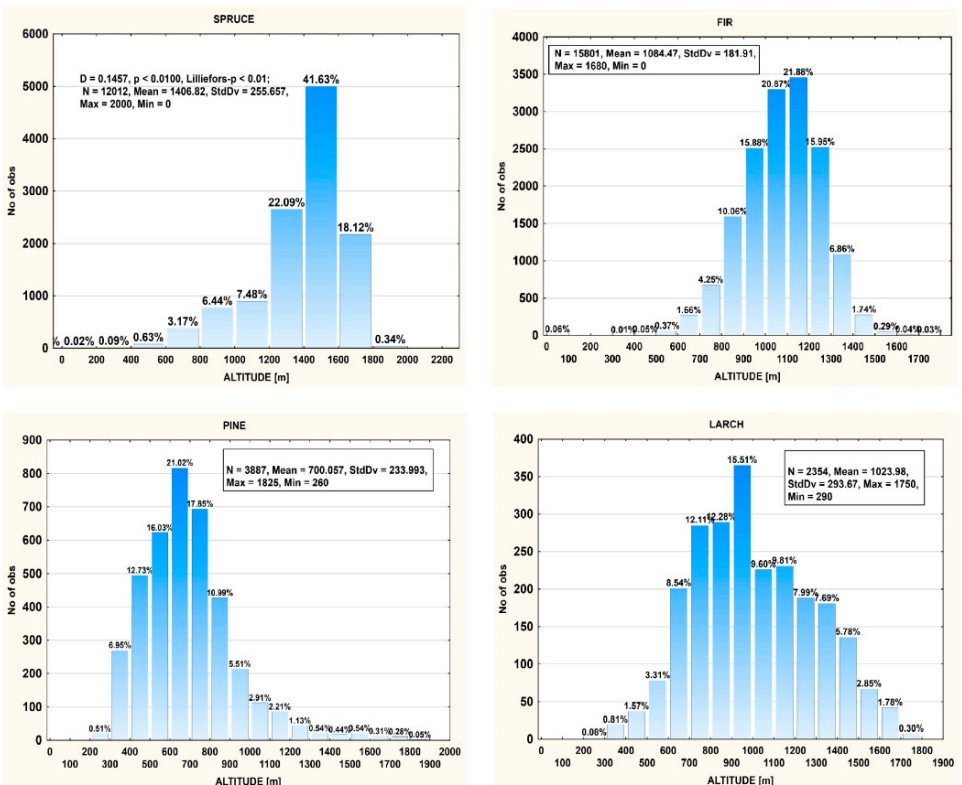

**Figure 2.** Distribution of stand elements based on altitude for resinous species located in Southern Carpathians.

Species distribution is very similar in terms of field inclination. The majority of species are situated on fields with a 30–40° slope (Figure 3).

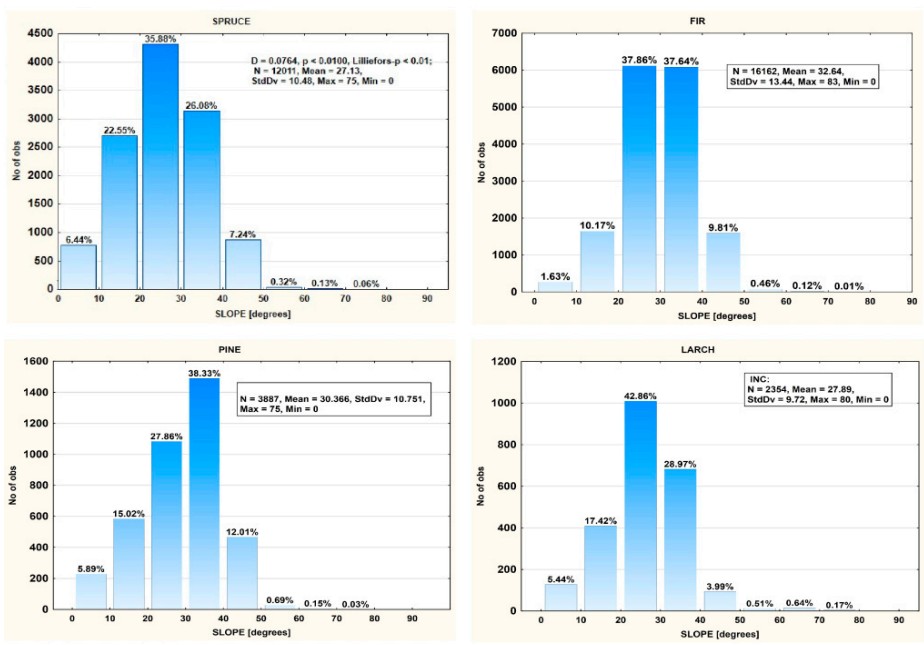

**Figure 3.** Distribution of stand elements based on field slope for resinous species located in Southern Carpathians.

Fir is distributed on shadowed field aspect, pine is located on sunny field aspect, whereas spruce and larch show no relationship to this parameter (Figure 4).

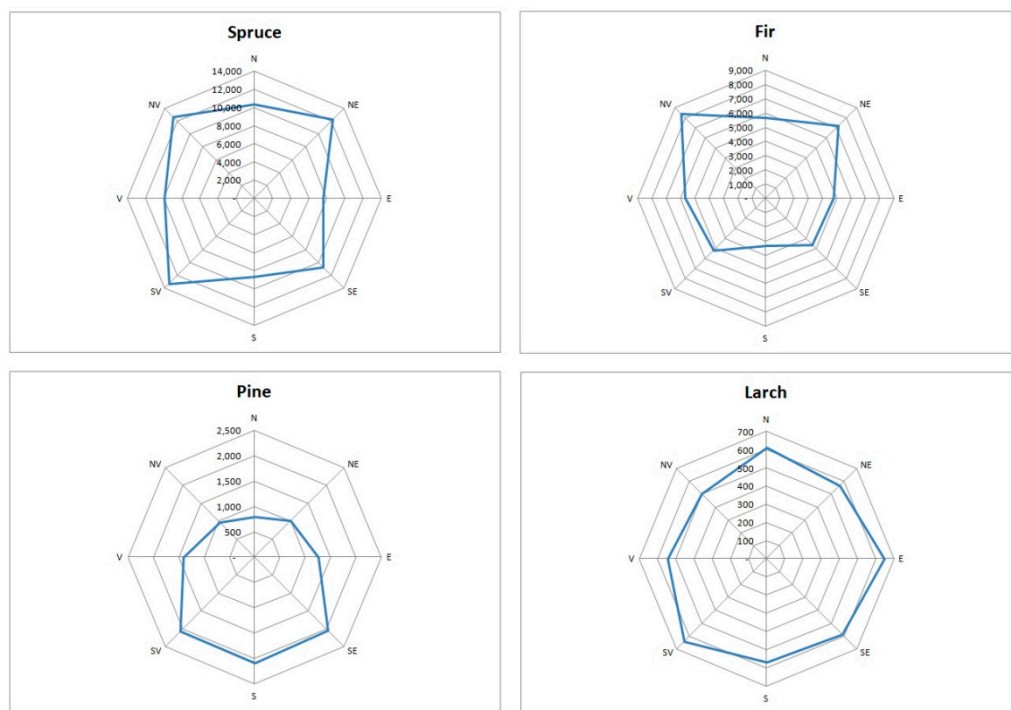

**Figure 4.** Distribution of stand elements based on field aspect for resinous species located in Southern Carpathians.

Fir has the oldest ages (with a high percentage between 80 and 140 years), while larch, pine, and even spruce have a majority of specimens with young ages (10–40 years) (Figure 5).

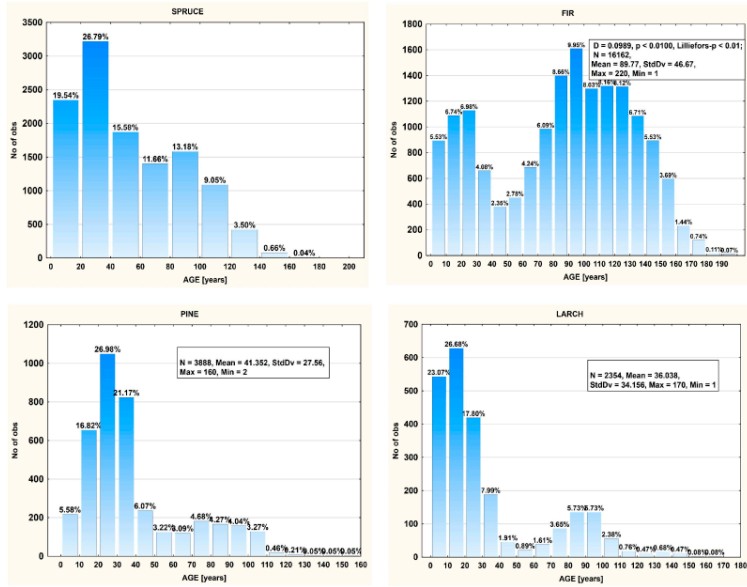

**Figure 5.** Distribution of stand elements based on age for resinous species located in Southern Carpathians.

Almost all species record a decreasing exponential curve in regard to the volume per hectare, with only spruce showing a slightly different pattern (Figure 6).

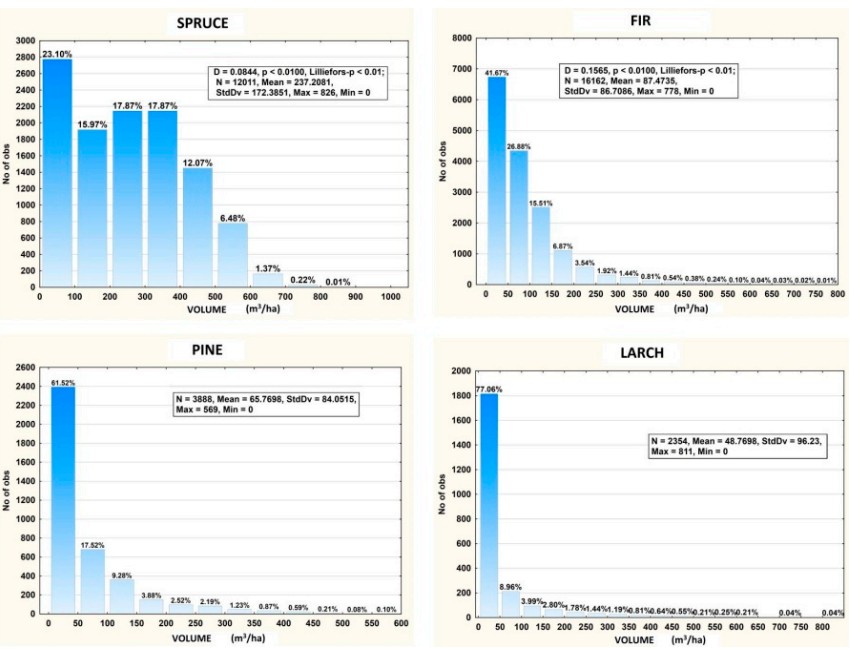

**Figure 6.** Distribution of stand elements based on volume for resinous species located in Southern Carpathians.

### 3.2. The Multivariate Analysis of the DataSet (Grouping Parameters)

Methods of multivariate data analysis were applied in order to identify more precisely the groups that influence each other as well as the grouping method for the measured parameters.

#### 3.2.1. The Principal Component Analysis Method (PCA)

This method allowed us to 'separate' the influences of dataset parameters in a number of groups: these are the main factors that explain the calculated variance.

Table S1 from Supplementary Material presents the results of the PCA analysis for each species, considering all 13 parameters included in this analysis. Thirteen main factors were identified for each species. If the first four factors explain approximately 60% for all cases, the last three factors cover only 3%.

The following analysis focused on studying the main four factors (Table S2 from Supplementary Material).

In order to keep the discussion short, we have analysed only the first four main factors here. For spruce, factor 1 is responsible for 27.95% of the total variance, considering the following parameters: INC, ALT, FLR, TS, and TP. Positive correlations exist between FLR, TS, and TP (Figure 7). This factor can thus be considered a specific factor—local geography.

Within factor 2, which is responsible for 16.01% of the total variance, EXP and RLF can be considered as parameters (as they correlate positively with each other), followed by SOL and CNF (which correlate negatively) (Figure 7).

Within factor 3, which is responsible for 13.56% of the total variance, VOL and VRT can be considered as parameters that are negatively correlated with each other (Figure 7).

Within factor 4, whichis responsible for 9.40% of the total variance, we can consider all the other parameters that were not included in the discussion.

For fir, factor 1 is responsible for 19.15% of the total variance and includes the following parameters: VRT and STR, which are positively correlated (Figure 7), followed by TS, TP, and CNS. We can name this specific factor—intrinsic properties.

Within factor 2, which is responsible for 13.83% of the total variance, FLR (which correlates positively) and ALT (which correlates negatively) can be considered as parameters (Figure 7).

Within factor 3, which is responsible for 12.10% of the total variance, we can consider the following parameters: VOL and CNF, and INC and SOL, which are negatively corre-

lated (Figure 8). We could name this factor a local specific property as it includes slope and soil properties.

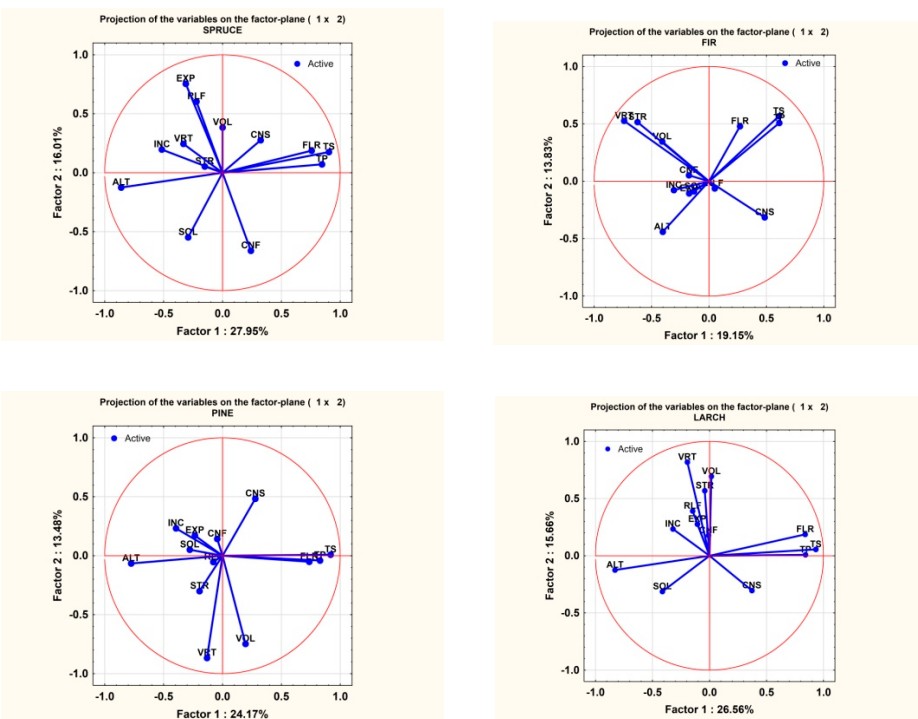

**Figure 7.** PCA diagram for factors 1 and 2.

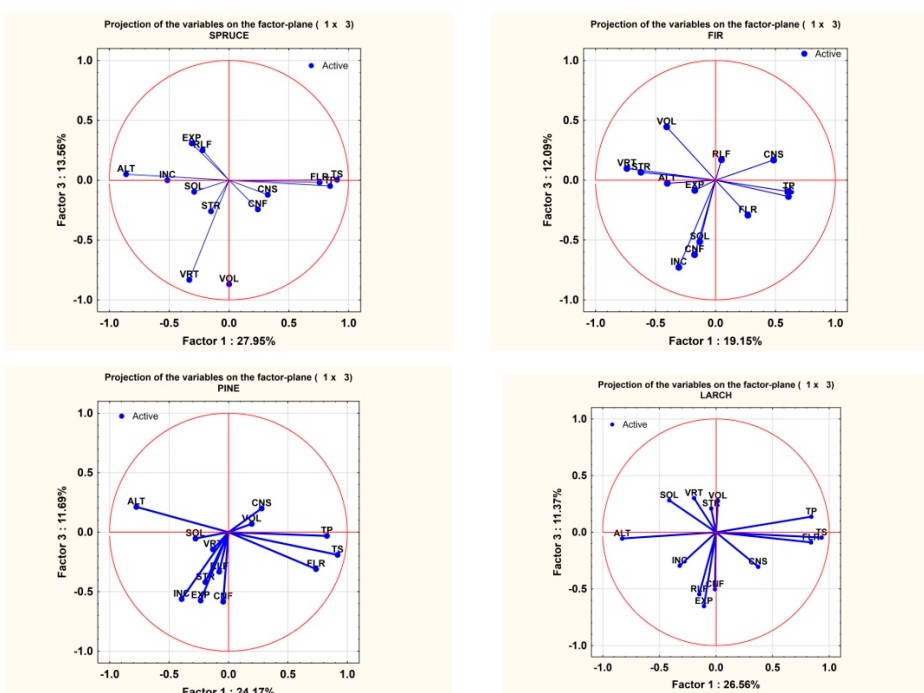

**Figure 8.** PCA diagram for factors 1 and 3.

Within factor 4, which is responsible for 8.74% of the total variance, RLP and EXP can be considered as parameters.

For pine, factor 1 is responsible for 24.17% of the total variance and includes ALT, FLR, and TP as parameters. These are positively correlated with each other (Figure 7).

Within factor 2, which is responsible for 13.47% of the total variance, we can consider the parameters VOL and VRT, which correlate negatively, and CNS, which correlates positively (Figure 7).

Within factor 3, which is responsible for 11.69% of the total variance, CNF, EXP, and INC (correlating negatively with each other) can be considered as parameters (Figure 8).

Within factor 4, which is responsible for 10.57% of the total variance, we can consider all of the other parameters: RLF, SOL, and STR.

For larch, we can observe that factor 1 is responsible for 36.56% of the total variance and includes ALT, FLR, TS, TP, and CNS as parameters that are correlated with each other (Figure 7).

Within factor 2, which is responsible for 15.66% of the total variance, VRT, STR, and VOL can be considered as parameters that correlate positively with each other (Figure 7).

Within factor 3, which is responsible for 11.37% of the total variance, RLF, CNF, and EXP can be considered as parameters (Figure 8).

Within factor 4, which is responsible for 8.88% of the total variance, INC and SOL can be considered as parameters.

Based on this method, we were able to separately discuss the following groups as parameters:

1-VOL–VRT–ALT
2-VOL–VRT–CNS
3-VOL–ALT–CNS

### 3.2.2. Factor Analysis Method (FA)

The FA method was applied to obtain a more complete and holistic image of data variation. The procedure allowed us to group the factors into four main categories. Table S3 presents the percentages for each factor. A very good correlation can be observed with the percentages of groups calculated through the PCA method (Table S3 from Supplementary Material).

Before making specific analyses, we must emphasize the existence of a sufficiently well-highlighted correspondence between the results obtained using the PCA method and the FA method, respectively. There are differences between the two categories of results, but, mainly, there is a 'beautiful' grouping between the 13 abiotic parameters studied and a preservation of positive and negative correlations between variables, which emphasizes the deep nature of the results obtained. There have been some differences spotted regarding the weights of the groups identified in the explanation of the total variant. Nevertheless, one can easily notice a coherence of the sets of results.

The percentages of each measure were calculated within each main factor. Significant similar groupings were obtained through the above method, with extremely rare exceptions (Table S4 from Supplementary Material).

For spruce, FA factor 1 is responsible for 27.95% of the total variance, and FLR, ALT, TS, and TP can be considered as parameters (Figure 9). This factor generally includes measures that were identified by the PCA method.

Within factor 2, which is responsible for 16.01% of the total variance, EXP, SOL, and RLF (positively correlated with each other) and CNF can be considered as parameters (Figure 9).

Within factor 3, which is responsible for 13.55% of the total variance, the previously identified parameters VRT and VOL can be considered (Figure 10).

Within factor 4, which is responsible for 9.40% of the total variance, we can consider all the other parameters, especially STR, INC, and CNS (Figure 11).

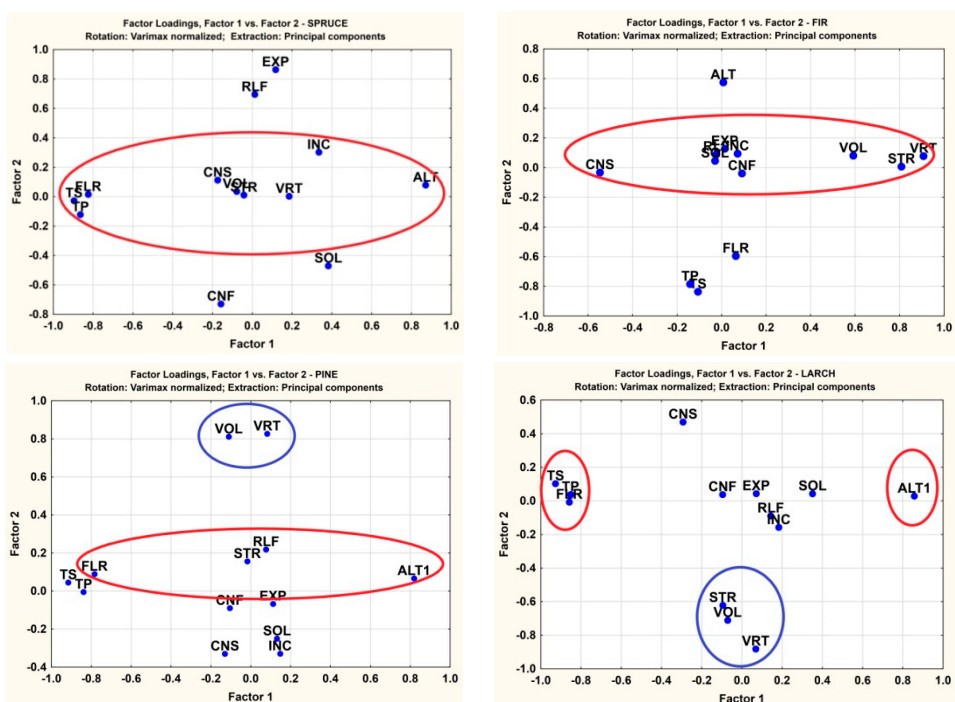

**Figure 9.** FA diagram for factors 1 and 2.

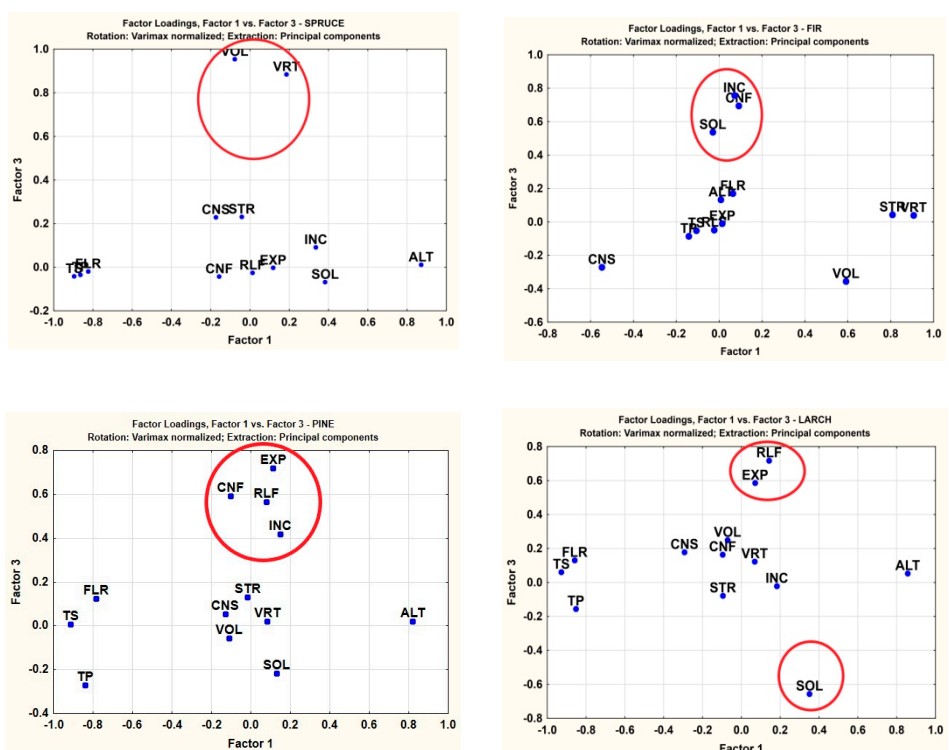

**Figure 10.** FA diagram for factors 1 and 3.

For fir, FA factor 1 is responsible for 19.15% of the total variance and includes the parameters identified already by the PCA method, namely VOL, VRT, STR, and CNS (Figure 9).

Within factor 2, which is responsible for 13.83% of the total variance, we can consider TS, TP, FLR, and ALT as parameters (Figure 9).

Within factor 3, which is responsible for 12.09% of the total variance, CNF, INC, and SOL (which are positively correlated which each other) can be considered as parameters (Figure 10).

Within factor 4, which is responsible for 8.74% of the total variance, RLF and EXP can be considered as parameters (Figure 11).

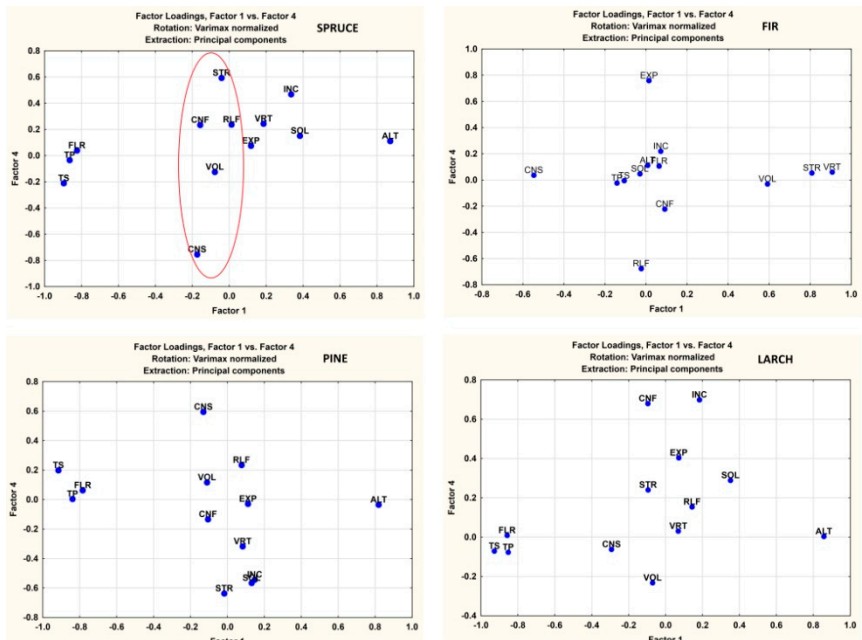

**Figure 11.** FA diagram for factors 1 and 4.

For pine, FA factor 1 is responsible for 24.17% of the total variance and ALT, FLR, TS, and TP can be considered as parameters. The last three are positively correlated with each other (Figure 9).

Within factor 2, which is responsible for 13.47% of the total variance, VOL and VRT (positively correlated) can be considered as parameters (Figure 9).

Within factor 3, which is responsible for 11.70% of the total variance, CNF, EXP, and RLF (which are correlated positively with each other) can be considered as parameters (Figure 10).

Within factor 4, which is responsible for 10.58% of the total variance, we can consider all the other parameters not involved, namely INC, SOL, and STR (negatively correlated) and CNS (negatively correlated) (Figure 11).

For larch, FA factor 1 is responsible for 26.56% of the total variance and includes ALT, FLR, TS, and TP as parameters that are positively correlated with each other (Figure 9).

Within factor 2, which is responsible for 15.66% of the total variance, VRT, STR, and VOL (negatively correlated with each other) and CNS (positively correlated) can be considered as parameters (Figure 9).

Within factor 3, which is responsible for 11.37% of the total variance, SOL, RLF, and EXP can be considered as parameters (Figure 10).

Within factor 4, which is responsible for 8.88% of the total variance, INC and CNF can be considered as parameters (Figure 11).

The positive correlations between the recording parameters are mostly maintained within both PCA and FA methods, corresponding to a trend of synchronized variation. For example, inclination and altitude have the same influence and appear to be positively correlated in the vast majority of diagrams.

### 3.2.3. Cluster Method

If we consider the weighted pair-group average Euclidian metric, the change consists of adding volume (VOL) to this first cluster (Figure 12). A sequencing based on similarity degree, without taking into account the distance's absolute value by using the Euclidian metric and an unweighted pair-group centroid connection, can order these parameters in a similar relationship.

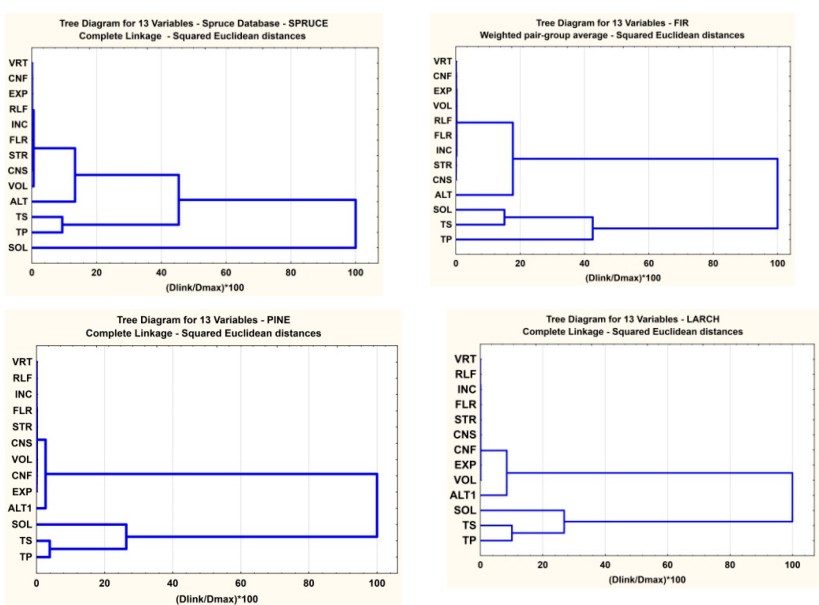

**Figure 12.** Cluster diagram: complete linkage squared Euclidean distances.

For spruce stand elements, we can observe a first cluster formed of RLF, INC, and FLR on one side (Figure 12) and STR, VOL, and CNS on the other side.

Fir shows a similar first cluster composed of RLF, INC, FLR, CNF, VRT, EXP, and VOL (Figure 12) and of STR and CNS. The results are similar if we consider the weighted pair-group average Euclidian metric (Figure 13).

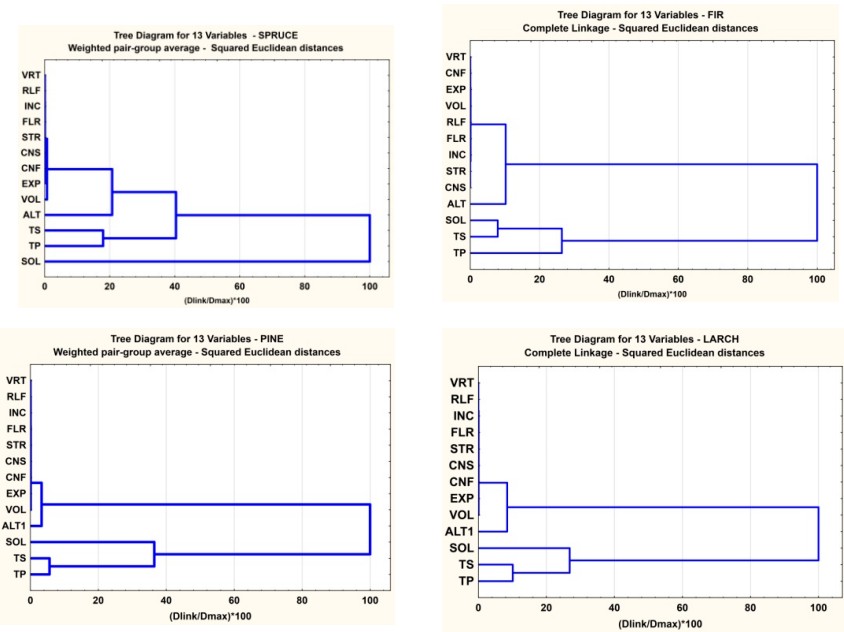

**Figure 13.** Cluster diagram: weighted pair-group average squared Euclidean distances.

For spruce stand elements, a particular group is composed of CNS and STR (Figure 14). Also, for fir, a particular group is composed of FLR, INC, STR, and RLF (Figure 14).

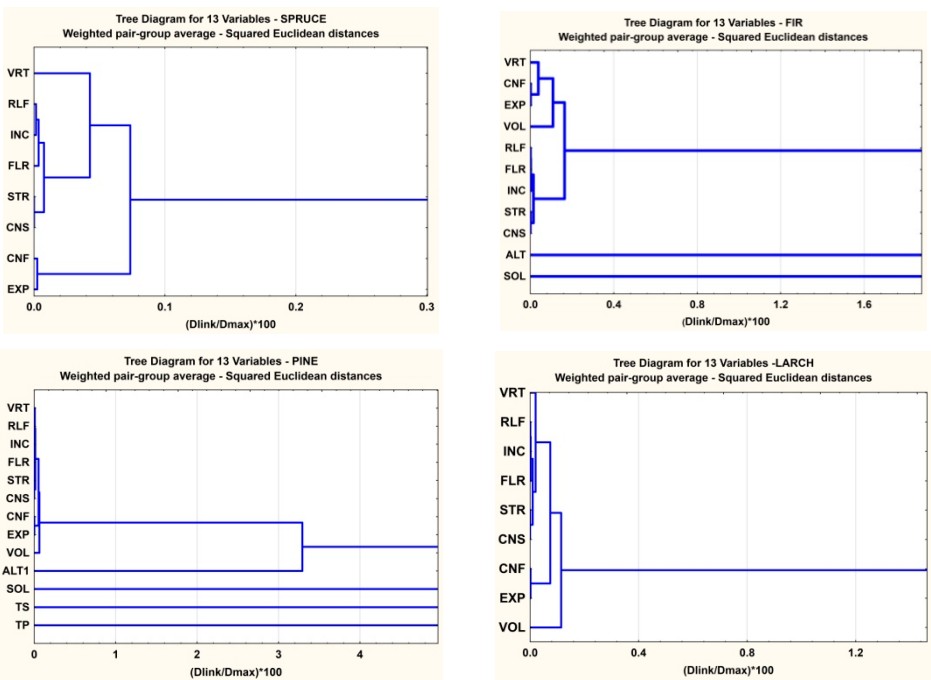

**Figure 14.** Cluster diagram: weighted pair-group average squared Euclidean distances—details.

Fir sequencing is ordered similarly to the spruce sequencing, with three small exceptions: INC and FLR have altered their places, and ALT has changed with SOL and TS in a circular permutation (Figure 15).

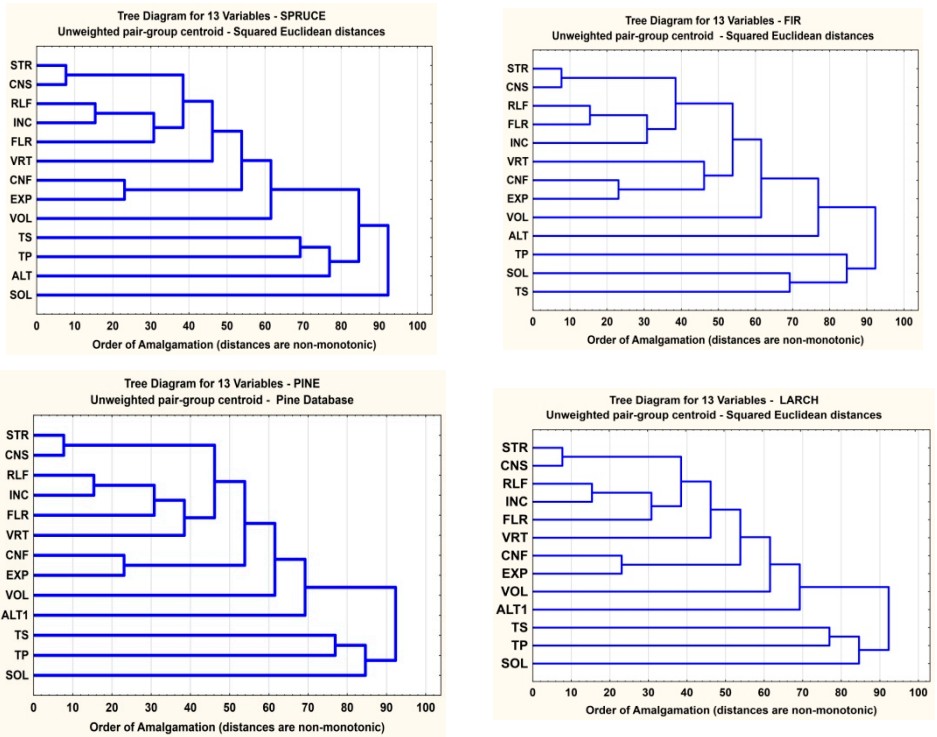

**Figure 15.** Cluster diagram: unweighted pair-group centroid.

Pine also records a first cluster composed of RLF, INC, FLR, CNF, VRT, EXP, and VOL (Figure 12). A particular group is composed of CNF, EXP, STR, and CNS (Figure 14). The results are similar if we consider the weighted pair-group average Euclidian metric. This sequencing is ordered similarly to the spruce sequencing (Figure 15), with a small exception: ALT and SOL have a circular permutation (Figure 15).

Larch (Figure 15) also shows a first cluster composed of RLF, INC, FLR, CNF, VRT, EXP, and VOL (Figure 13), very similar to the pine distribution. A particular group is composed of CNF, EXP, STR, and CNS (Figure 14)—exactly as in the case of pine distribution.

### 3.2.4. Three Dimensional (3D) PLOT Method

Figure 16 presents the VOL dependency, based on VRT and ALT, in the shape of a 3D surface.

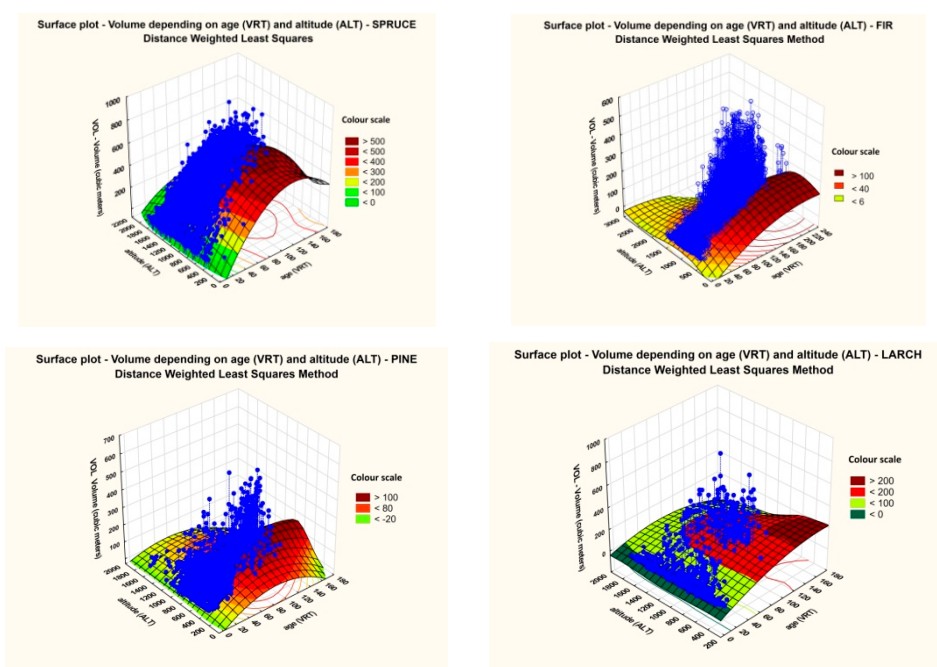

**Figure 16.** Three-dimensional VOL diagram based on VRT and ALT.

The diagram from Figure 16 represents the dependencies between volume (pictured on the vertical axis), altitude (ALT, meters), and age (VRT, years).

Spruce (Figure 16) presents a maximum local unstable surface, with a maximum age of approximately 120 years and an altitude of approximately 1000 m.

Fir (Figure 16) presents a maximum local unstable surface, with a maximum age of approximately 180–200 years and a low altitude.

Pine (Figure 16) presents a maximum local instable surface, with a maximum age of approximately 100–120 years and an altitude of approximately 800 m.

For larch, the considerations are similar to those for fir (Figure 16).

Figure 17 presents the VOL dependency, based on VRT and CNS, in the shape of a 3D surface. The diagram in Figure 17 represents the dependencies between volume (pictured on the vertical axis), consistency (CNS, percentages), and age (VRT, years). Spruce (Figure 17) presents a maximum local unstable surface, with a maximum for CNS and an advanced age, resembling the shape of a classical growth curve.

Fir (Figure 17) presents a similar shape, with a maximum for CNS and an advanced age, resembling the shape of a classical growth curve. However, the values are lower than in the previous case.

Pine (Figure 17) and larch (Figure 17) present maximum local unstable surfaces, with a maximum for CNS and an age of 100 years.

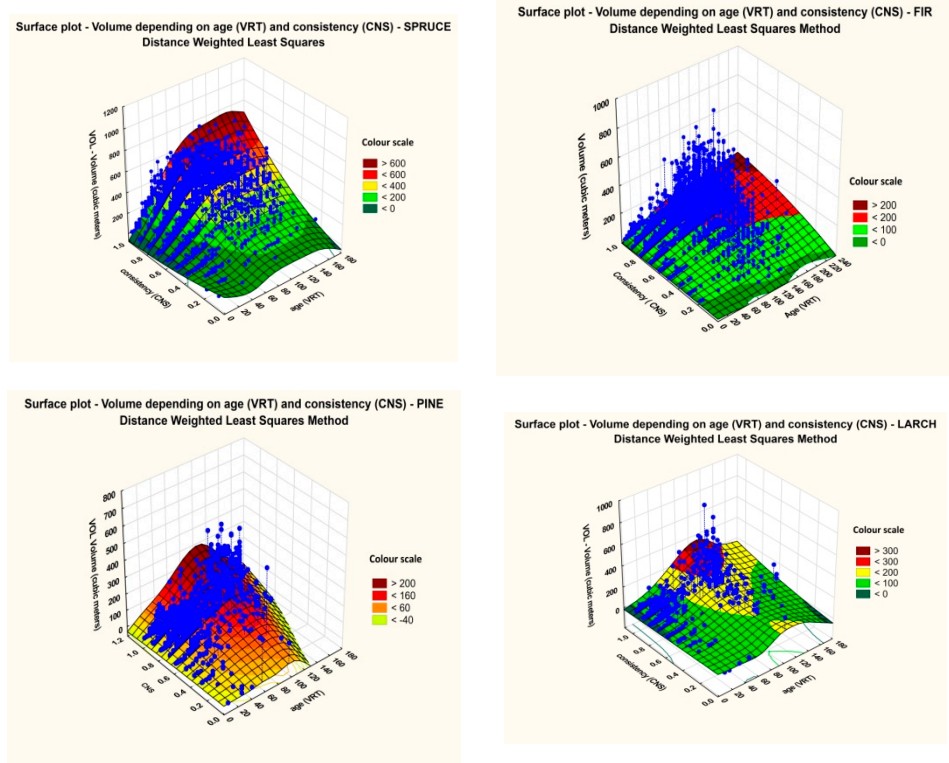

**Figure 17.** Three-dimensional VOL diagram based on VRT and CNS.

The diagram in Figure 18 represents the dependencies between volume (pictured on the vertical axis), consistency (CNS, percentages), and altitude (ALT, meters).

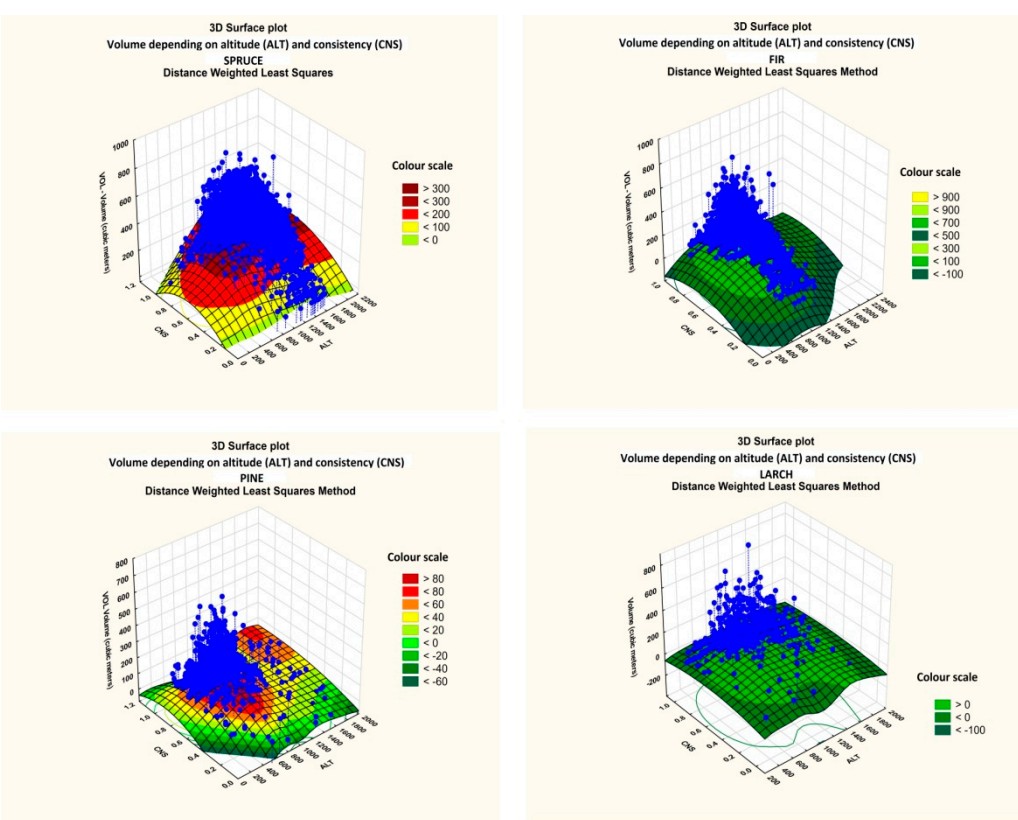

**Figure 18.** Three-dimensional VOL diagram based on ALT and CNS.

Spruce (Figure 18) presents a maximum local unstable surface, with a maximum for CNS (around 0.8) and an altitude of about 1000 m.

Fir (Figure 18) presents a maximum local unstable surface, with a maximum for CNS (around 0.6) and an altitude between 1000 and 1200 m.

Pine (Figure 18) presents two local maxima of maximum local unstable surfaces, with a maximum for CNS (around 0.7) and an altitude of about 700 m, and another maximum for high consistency and an altitude of over 1000 m.

Larch (Figure 18) presents a surface that implies that this species is not very sensitive to altitude and consistency.

## 4. Discussion

Concerning the distribution of stand elements based on age, we can observe a decrease in their number between 30 and 70 years. The silvicultural management applied concretely through the most recent thinning efforts explains this fact.

We can also observe that the characteristic grouping is common for the first two factors while using the cluster method for all studied stand elements. From this perspective, we can consider structure, consistency, relief, slope, and flora type as the most important influences for modelling the structure of the studied resinous stands. Furthermore, factors 3 and 4 do not record differences between the studied resinous stands. The only exception is the ALT parameter that shows changes in its percentage and shifts towards the cluster that contains TS, PS, and SOL.

PCA and FA methods showed that the strongest parameters affecting the structure of spruce stands are factors 1 and 2, namely altitude, flora, station type, and forest type.

The influence of the studied parameters on spruce stand structures is the same for both methods, with the exception of 'field slope'. This parameter has a higher percentage in PCA, appearing within factor 1, and a lower percentage in FA, where it appears in the last factor together with STR and CNS.

We can now group the 13 parameters that proved to be essential in influencing the evolution of certain resinous species in four big categories:

(a) A common factor for all species that includes a factor that groups age (VRT) and volume (VOL). It seems that age is the most important parameter for the increase of wood mass volume. This factor would correspond to intrinsic conditions.

(b) A common factor for all species that includes a factor that groups altitude (ALT), flora (FLR), TS, and TP. This factor would correspond to general location conditions.

(c) A common factor for all species that includes field slope (INC).

As a general observation, consistency (CNS) is an important factor for certain species when we speak about the level of wood mass volume. On the other hand, the influence of relief (RLF) is observably more diminished.

Some differences were discovered in another article for the 'slope' parameter in nine pure spruce plots from north-western Romania, in 'Valea Ierii'. The results of Plesa et al. 2017 showed that trees with a south-western field aspect have accumulated the largest amount of biomass, showing significant differences from trees on plots with north-eastern field aspect situated at the same altitude [33]. Field aspect was considered the principal factor regulating the forest's landscape function. In the entire set of analysed populations, the response function varied considerably within the south-western field aspect plots when compared to the north-eastern field aspect plots, but without significant differences related to tree density and altitude. All studied stands were pure and composed of even-aged spruce trees. The differences may be related to a range of factors such as altitude, field aspect, density, and other local conditions. A better growth of trees from the south-western field aspect was explained by the spruce trees' young stage and by their preferences for sunny and dry conditions [33].

Another experiment was conducted in the Curvature Carpathians in order to describe the effect of slope and field aspect on the received solar energy by using a digital elevation model. Some strong correlations were also found between radiant energy and

monthly average temperature on five plots installed at different altitudes on the eastern field aspect [34].

Volume and current volume increment were higher for silver fir on the northern slopes of the Southern Carpathians, while insignificant differences were recorded between the two slopes for Norway spruce [35].

Both methods have shown that fir stands are influenced by common factors such as VRT, STR, and CNS. In addition, differences between the two methods appear within the stand elements from the same species, with parameters such as TS, TP, and VOL varying in percentage.

In regard to stand age, in subtropical forests from China, scientists have found that the increase in tree size with species richness and stand age had a significantly positive effect on forest productivity [36].

Pine stands also showed that the first factor (which influences more than 20% of the stand's structure modelling) presents the following common characteristics: ALT, FLR, TS, and TP. All of the other factors (factors 2–4) have two common characteristics that influence structure at approximately the same level.

As a physical-geographic factor, altitude acts as an indirect primary factor that exerts an influence by means of climate [37].

A study carried out in forests from southern Anatolia in Turkey shows that the main abiotic determinants of vegetation communities are altitude and field aspect. These variables directly determine the site suitability for *P. brutia*, *P. nigra*, *A. cilicica* and *Cedrus-libani* [38].

For larch trees, the first two factors that influence their structure modelling have a percentage higher than 40% and have common characteristics (ALT, FLR, TS, and TP).

By analysing the differences between spruce and fir stand elements through PCA and FA methods, we can observe that STR and CNS parameters show high percentages for fir, as they are situated within factor 1, responsible for 10% of the total variance. On the other hand, spruce records a lower percentage, being situated within factor 4 that is responsible for 9.40% of the total variance. Therefore, it seems that CNS, which is also responsible for canopy cover, influences the structure of fir and larch stands to a greater degree than pine stands and even more so compared with spruce.

At the same time, field slope is more important for spruce than for fir, pine, and larch. This result is supported by the fact that pine is situated at the lowest altitudes, and is used especially on degraded lands and in afforestation formulas for this type of field [19].

The SOL parameter influences the structure of fir stands more than pine or larch. It is known that Romanian fir is a species more affected by climatic–edaphic conditions [39]. At the same time, the difference between the SOL influence for spruce and fir can be explained by spruce's lateral spreading roots, while fir has stronger anchoring in soil due to its top and lateral spreading roots.

In the Austrian basal area increment model for individual trees, topographic factors like elevation, slope, and aspect explained up to 3% of the variation, as did soil factors. The remaining site factors such as vegetation type and growth district accounted for a maximum of 3% of the increment variation. In total, site factors explained from 2 to 6% of the increment variation [40].

In other regions, such as the Amazon basin, researchers have discovered strong evidence to support the fact that chemical and physical soil properties are the main factors that determine variations in forest structure and dynamics [41].

It also seems that relief is the parameter that orders resinous species from the most important towards the least important one, as follows: spruce, larch, pine, fir.

In regard to the field's orography, the RLF factor influences the spruce stand structure by a higher percentage, being grouped within factor 2, while it appears in factor 3 for larch, in factors 3 and 4 for pine, and in factor 4 for fir.

The two VOL and VRT parameters are grouped for most species, as age directly influences stand volume. For almost all species, these parameters are found within factor 2 that is responsible for approximately 15% of the total variance.

EXP, the factor responsible for field aspectis found within factor 3 for larch and pine in all three applied methods. The difference consists in the fact that PCA and FA place it within factor 2 for spruce stands and within factor 4 for fir stands.

A study from central Italy examined the relationship between site index and environmental factors in Douglas-fir plantations in the province of Firenze. Approximately 58% of the observed site index variation was explained by annual rainfall, water surplus, clay content, calcium-carbonate content, and east/west aspect component. Climatic factors have a greater influence on the productivity of examined Douglas fir plantations than examined topographic and soil factors [42].

Three common characteristics (ALT, TS, and TP) affect the modelling of all resinous stands studied in the Southern Carpathians. Altitude, station type, and forest type are the common characteristics that are found with both PCA and FA methods and are the main influences responsible for the spread and structure of stands composed of the studied resinous species (spruce, fir, pine, larch).

## 5. Conclusions

From an altitudinal perspective, pine, larch, fir, and spruce are ordered ascendingly, starting from the Getic Piedmonts (high plateaus in the southern part of the Southern Carpathians). They start with a minimum altitudinal interval of 400 m (for pine stands) and reach a maximum of 1800 m (for spruce stands). The majority of resinous stands vegetate on fields with a 20–40° slope. Spruce and larch stands are present on most field aspects. The same cannot be said about fir, which prefers shadowed field slopes, or about pine, which prefers sunny field slopes.

A set of common characteristics have resulted from our study that focused on characteristics measured during 1980–2005 over a total surface of 143,431 hectares in the Romanian Southern Carpathians. These common characteristics have been found by applying PCA and FA methods, and are represented by altitude (ALT), station type (TS), and forest type (TP). They influence the spreading and structure of the studied resinous stands to a higher degree, regardless of species. The percentages calculated for these characteristics actually represent the percentages of the influences of these abiotic factors in modelling the structure of resinous stands from the Southern Carpathians.

The cluster method indicated two other characteristics that are common for all resinous stands and that strongly influence the modelling of their structure. These characteristics are structure type (STR) and consistency type (CNS). Their percentages obtained through the cluster method are determined more by the application of forest management plans.

Station type and forest type are fundamental factors in modelling the structure of resinous standslocated in the Southern Carpathians.

Taking into account the fact that the studied forests are growing under forest management, the species that have formed the main stand elements were chosen so that the optimum quality would be obtained. This is established based on the presence and influence of forest ecosystem components and should be strongly connected with a certain type of vegetation.

Overall, the findings of the present study provide a useful theoretical basis for sustainable forest management that supports the resilience of forests in the context of multiple challenges.

**Supplementary Materials:** The following are available online at https://www.mdpi.com/article/10.3390/f12081029/s1, Table S1. Cumulative variance explanation for the first 13 Eigen values for each species database. Table S2. Variables factor coordinates, based on PCA method. Table S3. Principal components cumulative eigenvalues contributions. Table S4. Factor Loadings (Varimax normalized Method) Extraction: Principal components (Marked loadings are >0.70).

**Author Contributions:** Conceptualization, L.D. and G.M.; methodology, L.D., G.M.; software, D.M., G.D.M., M.D.D., B.R., G.M.; validation, G.M., D.M. and M.D.D.; formal analysis, B.R.; investigation, N.T., V.C.; resources, L.D.; data curation, D.M., M.D.D., B.R.; writing—original draft preparation, V.C., N.T.; writing—review and editing, L.D.; visualization, N.T., V.C., L.D.; supervision, L.D., G.M., L.G.; project administration, L.D.; funding acquisition, L.G. All authors have read and agreed to the published version of the manuscript.

**Funding:** This work was supported by the project "Excellence, performance and competitiveness in the Research, Development and Innovation activities at "Dunarea de Jos" University of Galati", acronym "EXPERT", financed by the Romanian Ministry of Research and Innovationin the framework of Programme 1—Development of the national research and development system, Sub-programme 1.2—Institutional Performance—Projects for financing excellence in Research, Development and Innovation, contract No. 14PFE/17.10.2018. The work of Rosu Bogdan was supported by the project "ANTREPRENORDOC", Contract No. 36355/23.05.2019, financed by The Human Capital Operational Programme 2014–2020 (POCU), Romania.

**Data Availability Statement:** Datasets generated and/or analysed during the current study are available from the corresponding authors on request.

**Acknowledgments:** We would like to thank the editor and anonymous reviewers for their useful advice that helped to improve the manuscript.

**Conflicts of Interest:** The authors declare no conflict of interest.

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
