# Peer review of "Structural Characteristics of the Main Resinous Stands from Southern Carpathians, Romania"

_forests, doi:10.3390/f12081029_

Round 1

Reviewer 1 Report

Dear Authors, the paper 'Structural characteristics of the main resinous stands from Southern Carpathians, Romania ' represents an interesting approach to a well known topic. I believe the Authors defined the nische for the research appropriately and the paper may be a contributionn to the field.

The abstract represents the paper accurately and gives a proof of results and conclusions obtained in the paper.

The Introduction provides sufficient background on the topic and the refference list is up to date. The introduction should be ended with an overall sentense regarding the carried research, at the poiunt in only informes about statictical analysis applied in the results section.

The methods are well described, the area is well defined. It would be valuable if the authors not anly mentioned the methods they applied, but also expleined the reasons for this choice. This way it would be possible to compare the results with other in the field, with respect to the same motivation of other authors. 

REsults and modelling are presented clearly. However they are based on single run, therefore it seems control is missing. The discussion is supported by the results. Similar approaches from the literarure are discussed as background.

The choice of factors on diagram 9 isa not clear for the reviewer. Also the interpretiation for positive and negative correlations are missing.

Is the range of experiment accurate? The values on figure 18 are very close to the border. The trend may not be representative.

The discussion and conclusions are drawn carefully and significant.

Some problems with citations occur , please re-check.

Reviewer 2 Report

The concept of study is interesting with different stages of statistical analysis, study brings some methodological novelty, findings are possible to generalize in the broader management efforts beyond regional importance.

Commnents and suggestions for authors: 

  • Page 3, line 102. The objective of the paper is formulated as assessment of the impact of abiotic factors on forest structures, but literature review in introduction is aimed mostly to the general information about tree species composition in Romanian Carpathians Mount. I would recommend to the authors to include to review also specific titles connected more closely to the study objectives, or more detailly specify in objective section to which abiotic factors the study concerns.
  • Page 4, lines 132–136, sampling design is a bit unclear, reasoning of 2-3 sample plots, which type of sampling was applied, how average d.b.h. was calculated, what are representative trees for height measurements, measurements of d.b.h. and height with tolerance ±10% means accuracy? Ultrasonic or laser hypsometers accuracy is about ±1%. And also, in the beginning of chapter 2 is statement, that data were collected from forest management plans, than diameter and height measurements is a part of what?, or it is only description how were these variables, also with volume and age  collected during preparation of forest management plans in the past decades?  this section of materials and methods – page 4, lines 132–150 is unclear and could be chaotic for the readers. Therefore, I recommend rewrite it more clearly.
  • Some statistics in Figures 3,5,6 are interpreted by comic way, for instance mean altitude, slope, age and volume with four decimals, for variability interpretation is better CV%
  • Page 8, line 265 correctly should be “If the first four factors explain……”
  • Page 12, line 375, correctly should be 3.2.3
  • Figures in cluster analysis, Figures 12–15 have the same names, it is confusing, also which one is Figure 14a, or 15.b, 14.d? please mark it correctly, the same for figures in chapter 3.2.4
  • Page 14, line 408, correctly is 3.2.4
  • The scale of the most figures is too small, figures are unreadable, I am not sure if the histogram fillings were chosen well – lower part of histogram columns is fading, paper range is quite large – 24pages, some space is possible to save because some redundance exists between figures and appendix tables and maybe not all figures is necessary to publish in the paper.

Author Response

This manuscript is a resubmission of an earlier submission. The following is a list of the peer review reports and author responses from that submission.

Round 1

Reviewer 1 Report

FORESTS

Title Modelling forest structure of the main resinous stands from Southern Carpathians, Romania

General comments:

This paper addresses one key question in resinous stands growing in Southern Carpathians, Romania, in particular the role of abiotic factors on forest structure. Therefore, the topic and data included in the manuscript should be of interest to readers of Forests. Currently there are many very general sentences in the manuscript, which makes the novelty and implications of this paper more difficult to identify. Although several changes should be accomplished to improve the understanding of some parts of the text and also to clarify some of the methods followed, I consider that the manuscript flows well, and the materials and the methods are in line with the aims to be addressed. However, the “results” section is not presented in a coherent way according to the specifics objectives. Besides, in the discussion, some parts seem to be unsupported conjecture confusing association with causation. Authors need to support their ideas using citations and/or more careful language. Need further work. Based on those comments, I would recommend a major revision in which the comments and suggestions below are fully addressed. The following are suggestions that I hope authors may consider for each respective section of the manuscript. I hope that my comments will be helpful in a future version of the manuscript.

Comments and suggestions:

Highligths

  1. The generality of writing in the manuscript is also reflected in the highlights that include several references to methodology, but they are ambiguous respect the main results.

Key words

  1. Reconsider key words, some of them are mentioned in the title: Southern Carpathians.

Abstract

  1. The summary is too general, and in many cases not very specific. The authors must provide statistical support to their statements, including in the summary a minimum of numerical data.

Introduction

  1. In general, the introduction is badly written, besides the structure is a bit confusing.
  2. Generally, the description of the study area is included in the “materials and methods” section.

Material and methods

  1. Statistical analysis

Regarding statistical analysis, I suggest:

Descriptive statistics, improve redaction, particularly data normality and variables transformation

Correlations, simplify redaction and avoid redundancy in the methods description.

Results

  1. In concordance to the Mat&Met section, the Results section is also very confused, and it should be improve its clarity.

I recommend keeping the same order of result presentation that indicated for statistical analysis.

Discussion

  1. Compared to previous sections, the Discussion is the strongest section of the paper which contributes to improve general clarity of the paper.

Some parts seem to be unsupported conjecture confusing association with causation. Authors need to support their ideas using citations and/or more careful language. Need further work.

Divide the discussion section into subsections (by specific objectives) = facilitating the reading and clarity of the discussion

The authors should be more careful with the selection of bibliographic support, looking for a consistency between the statistical approach of the study, and scientific support using to justify their results.

Conclusions

  1. I think that conclusions are too general and obvious. Conclusions should emphasis more consideration about your conceptual proposal. All this section must be rewritten. Keep it clear and make sure you are really proposing something new. The section is very “wordy”, please bring out the main ideas and simplify redaction.

Tables and figures

  1. Tables 1 - 4 – These tables are not necessary. Move to supplementary material.
  2. Figure 2, 3, 5 and 6. – The information contained in these figures would be better contained in a table.

32.- Specific comments

English must be improved. I encourage the authors to have their MS reviewed by a colleague whose mother language is English.

Reviewer 2 Report

The work covers the research problem posed and correctly relates to the source literature. The cited literature could be supplemented with the latest items. The terminology used is correct. The presented results constitute a complete set of data, exhaustive criteria for comparability. The discussion and the presented conclusions are supported by the experimental part. The work structure, style and linguistic correctness are a strong point of the work. 

The analyzes were carefully planned. The obtained results constitute a complete description of the given problems within the defined range. The conclusions formulated in the work are supported by the obtained results. The presented data set makes it possible to determine the features of the tested solutions and their operational parameters. 

The technical equipment and the scope of its application have been described correctly. The methods used were correctly discussed and laboratory procedures for sample preparation and results processing were presented. The specialized tools are freely used in the implementation of tasks resulting from the preparation of the paper.

Overall, the presented data and the discussion are carried carefully and only a few gramatical errors might be corrected.

THe Authors should also consider an expansion of the reference list, the paper should be exposed to a wide reader.

Reviewer 3 Report

Dear Authors,

I have finished my review on your manuscript. I am sorry to conclude that the paper should be rejected in this version due to many issues/reasons. Please refer to my detailed comments.

Best regards,

R.

Title: It is unclear what the title stands for.

Abstract: Needs improvement. First of all, the authors should think about giving a background for their research and formulating the problem to be solved. Then the methods used and the main results and conclusions can be formulated.

Line 16: what the authors mean by “The data were collected from 1980 until 2005…”? Were they collecting this data? Was the data intended as a time-series analysis?

Line 18: what are the “lines” ?

Line 20: be more specific about what abiotic factors were evaluated and what kind of impact.

Line 25: what is a “station”?

Introduction

General comment: it is poorly written, has a lot of grammar issues, and fails to characterize the problem pursued by the authors. Also, it seems that the authors need to improve their vocabulary on the forestry terms. The importance of their research is far away of being evident from what they have given here, which is merely a description of the Romanian forests and of the species they are targeting by their study. The aim of the study is rather ambiguous since no problem to be solved has been identified by the given introduction. Citing system is uncommon and does not adhere to the template and instructions for authors.

Specific comments:

Line 31: separate “relateddata”

Line 39: requiresempirical ???

Line 41: relieson

Lines 42-43: give some references here.

Line 45: tothe

Lines 43-44: to what are referring the authors by “complex mountain system”?

Figure 1: remove it from the introduction.

Lines 52-57: the information does not belong to introduction. It is rather belonging to Materials and methods.

Line 57: give also the common name and give a reference for the statement from lines 57-59.

Lines 59-60: give common names also.

Line 61: to what are the authors referring by “smart forests”?

Line 62: Spruce has a protective role areas prone to erosion… I am not sure that I understand to what the authors are referring to. Check also the font type.

Line 64: I am not sure but I think that Picea abies has the common name of Norway spruce.

Line 76: Pinus genera?

Line 78: P. cembra, restricted to small populations ???

Line 78: what do the authors mean by “massifs”?

Line 86: what do the authors mean by “genetic”?

Line 89: what do the authors mean by “situated”?

Line 89: what do the authors mean by “skeletal”?

Materials and methods

General comment: materials and methods are highly unstructured, failing to give the necessary information for replicability of the study. I am unsure if the statistical approach is sound. I have my doubts about the techniques used also due to the introduction statements which failed to guide the reader to the problem. No hypotheses were formulated there to see if the statistical approach was sound. English and formulation of the text are poor in this section. No references are given to the software used and the type of data taken into analysis is not sufficiently and clearly described to judge what kind of statistical methods would fit.

Specific comments:

Line 103: To what are the authors referring by “realised”?

Line 106: Again, I am unsure what “lines” are… neither the elements of the trees…

Line 109: the authors need to be consistent. Here they are referring to Norway spruce. What are the “pure stands”?

Line 110: “we usedall the standsin” ???

Lines 114-117: Did the authors measured these diameters in the given time frame?

Lines 118-119: „trees enclosed at the average diameter”???? Not sure what this means...

Lines 119-120: „The age was determined for each stand, with a tolerance of +/- 10%, based on the forest management plan” Not sure what this means. Also, the authors refer commonly to „determination”. Determination is rather something mathematical.

Lines 118-126: So, the authors have been counted the growth rings of these species? Perhaps they could give some details on how the measurement was done and on how has been carried out the analysis in the laboratory. Were there any problems in counting the growth rings of these or part of these species?

Line 125: again, I am not sure what an “element” is.

Lines 127-133: It is quite unclear what procedures have been used. I am not sure if “expressed” is appropriate in this context. Some text needs checking for font type.

Lines 134-136: double-check the concepts and terminology.

Lines 134-142: double check. There are some formulations such as: geographic exposure (exposure to what?), flora characteristics (what characteristics? Height? Density? Species? Dominance? Abundance? ????), relief unit (????), field slope (????), field configuration (????, how can one configure the field?), tree structure (I think it is the stand structure), station type.

Line 149: perhaps the authors could clarify what is the “multi-varied methods of analysis of complementary results”, in particular multi-varied.

Line 151: to what are the authors referring by “extremely consistent”?

Line 155: the authors are repeatedly referring to the work of Popa. Perhaps they could explain clearer their statistical approach since these procedures are common in many sciences and cannot be attributed to a single paper.

Line 158: to what are the authors referring by “unitary”?

Lines 160-161: Thus, for complementarity, the correlation coefficients were also analysed by comparison with results obtained through the stepwise regression method... Correlation coefficients are typically computed by a correlation analysis and not by regression.

Lines 166-170: same problem

Results

General comment:

Many of the results given make no sense for interpretation since these are purely based on what the statistical analysis has returned. No interpretations are given to make it clear for the reader what the significance of the results would be or how should one take a look at them. It is unclear what the authors try to demonstrate. In my opinion, the results are purely a comparison of statistical methods run on the same dataset, not showing something revolutionary but a typical distribution of forests and association of factors. Paragraphs are rather uncommonly configured with only one sentence in each.

Specific comments:

Lines 201-204: this is common knowledge

Lines 209-201: to what this result helps? It is rather descriptive and a state of those species.

Lines 213-214: also, this….

Lines 225-227: are purely methods.

Tables 1-4: it is quite uncommon to report as results the eigen values and coordinates. These should be some supporting data.

Line 238: indeed, it is a good reason. ?

Line 248 and anywhere else: what is “responsible”?

Line 254 and anywhere else: Ox axis????

Discussion

General comment:

Discussion section seems to be highly unstructured and fails to guide the reader in understanding the results. I am not sure that authors have chosen the best approach to interpret their results. The same problems related to the construction of paragraphs, phrasing, terminology used and citing persist also here. By its formulation, the discussion section is rather leading nowhere.

Specific comments:

Lines 439-440: But wasn’t the study about the southern part of Romania?

Lines 440-443: Whose results? Yours?

Line 443: Slope exposure?

Line 444: What is a “response function”?

Line 446-447: Your stands?

Line 449: What is “superior growth”?

Lines 452-456: How this relates to your results? It is not clear to me…

Line 457: What is the “current volume increment”?

Line 460-461: this is common knowledge.

Lines 464-466: common knowledge…The higher the tree size the higher the productivity.

Lines 471-472: what is a “secondary periodic factor”?

Lines 483-484: what is “tree’s closing”?

Lines 490-492: So, what is new in your study?

Line 494: What is a “top spreading root”?

Line 500-502: I would say that the authors go too far by comparing their results to those from the Amazon given the contrasting growth conditions from there.

Conclusions

General comment:

Fail to conclude since the above-discussed sections are not OK.

Specific comments:

Lines 526-532: Isn’t this common knowledge? I assume that the management plans from which you have taken the data indicate something about this.

Lines 533-540: Have the authors run a time-series analysis? Or this is just a state of art based on data collected from several management plans developed at different time intervals?

Line 543: Aren’t these the effect of management?

Lines 546-547: Only in the Carpathians? How about other forests?

Line 550: to what is the “bonity” referring in this context?

Lines 553-555: I am not sure how…